# Deterministic Inference across Tensor Parallel Sizes That Eliminates Training–Inference Mismatch

**Ziyang Zhang** [* 1]  **Xinheng Ding** [* 2]  **Jiayi Yuan** [3]  **Rixin Liu** [3]  **Huizi Mao** [4]  **Jiarong Xing** [3]  **Zirui Liu** [2]

## Abstract

Deterministic inference is increasingly critical for Large language model (LLM) applications such as LLM-as-a-judge evaluation, multi-agent systems, and reinforcement learning (RL). However, existing LLM serving frameworks can produce different outputs for identical inputs when tensor parallel (TP) size or batch size changes, even under greedy decoding. This arises from the non-associativity of floating-point arithmetic and inconsistent reduction orders across GPUs. While prior work has addressed batch-size–related nondeterminism through batch-invariant kernels, determinism across different TP sizes remains an open problem, particularly in RL settings, where the training engine typically uses Fully Sharded Data Parallel (FSDP) (i.e., TP = 1) while the rollout engine relies on multi-GPU TP to maximize the inference throughput, creating a probability mismatch that can degrade or even collapse training. We identify and analyze the root causes of TP-induced inconsistency and propose **Tree-Based Invariant Kernels (TBIK)**, a set of custom matrix multiplication and reduction kernels that guarantee bit-wise identical results across TP sizes. Our key insight is to enforce a consistent reduction order across and within GPUs. We implement TBIK in Triton and integrate it into vLLM and FSDP, achieving **bit-wise deterministic inference** across different TP sizes and **zero probability divergence** between rollout and training engines in RL pipelines. By eliminating mismatches caused by different parallelization strategies, TBIK enables **true on-policy RL at scale for the first time**, leading to improved model performance and faster convergence.

---

[*]Equal contribution  [1]Independent Researcher [2]University of Minnesota, Minneapolis, Minnesota, USA [3]Rice University, Houston, Texas, USA [4]NVIDIA Corp., Santa Clara, California, USA. Correspondence to: Zirui Liu <zrliu@umn.edu>.

*Proceedings of the $43^{rd}$ International Conference on Machine Learning*, Seoul, South Korea. PMLR 306, 2026. Copyright 2026 by the author(s).

## 1. Introduction

Large language models (LLMs) are increasingly powering various real-world applications (Chang et al., 2024; Yuan et al., 2025b; Yan et al., 2025). In many of these scenarios, reproducibility is essential, meaning the generation process must be deterministic: given a fixed random seed and identical inputs, the model should produce the same outputs across different system configurations and different runs.

We emphasize that determinism is particularly crucial in three contexts. *(1) Evaluation with LLM-as-a-judge* (Zheng et al., 2023; Li et al., 2024; Thakur et al., 2025; Zhou et al., 2025; Zhang & Jing, 2026), where LLMs are used as evaluators to assess the quality of other models. If the judge model produces inconsistent evaluation for same inputs, the comparison becomes unreliable. *(2) Multi-agent systems*, where multiple LLMs collaboratively complete one task (Guo et al., 2024; Tian et al., 2025). The nondeterminism of one model can cascade across the entire system, producing divergent trajectories and making debugging challenging (Huang et al., 2025). *(3) Reinforcement Learning (RL) with LLMs* (Shao et al., 2024; Sheng et al., 2025; DeepSeek-AI, 2025), where training and rollout usually use different frameworks (e.g., Fully Sharded Data Parallel (FSDP) (Zhao et al., 2023) vs. vLLM (Kwon et al., 2023)) and configurations (e.g., different tensor-parallel (TP) sizes). As shown in Figure 1, the probabilities computed by the training and rollout engines exhibit a large gap, which can significantly affect the learning process and make an on-policy RL process behave more like off-policy (Yao et al., 2025b).

Unfortunately, prior studies have shown that determinism is a *missing* property in today's LLM systems. Even under greedy decoding, changing system configurations such as batch size and TP size (Yuan et al., 2025a; He & Lab, 2025) can cause the same input to produce different outputs, leading to up to 9% variation in accuracy on the AIME dataset (Yuan et al., 2025a).

The fundamental reason is that floating-point (FP) arithmetic is non-associative, so different computation orders may lead to different results due to rounding errors. In LLM serving systems, such variations arise from multiple sources, including continuous batching (Yu et al., 2022),

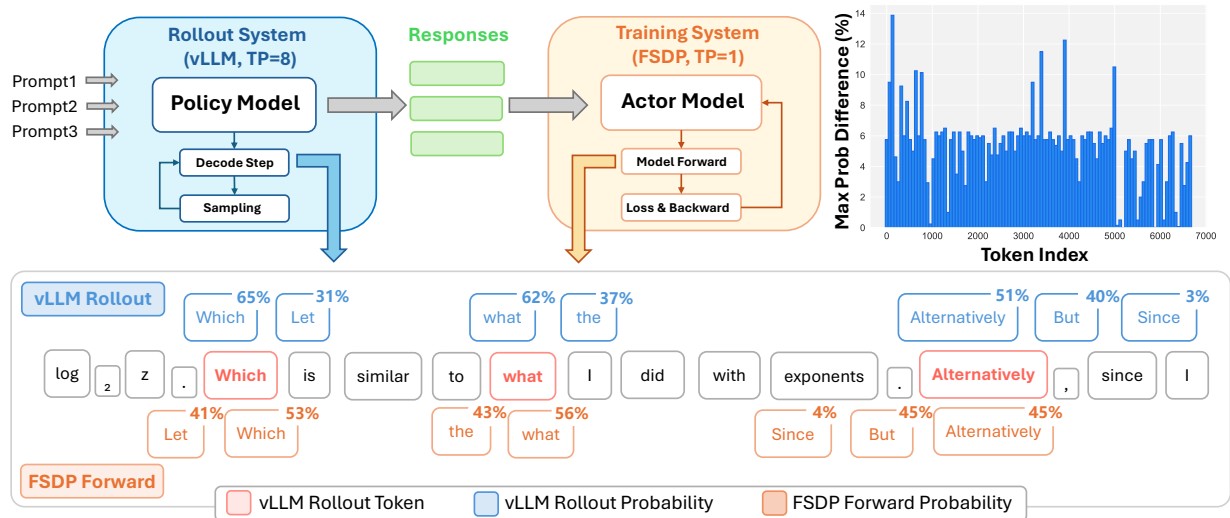

*Figure 1.* Illustration of probability discrepancies in RL training. Different frameworks and TP sizes produce noticeably probability gap for the same response, hindering stable and truly on-policy RL.

parallel strategies such as tensor parallelism, and different kernel implementations, etc.

To address this issue, (He & Lab, 2025) introduced *batch-invariant operations*, including batch-invariant FlexAttention (Dong et al., 2024), RMSNorm, and matrix multiplication (MatMul) kernels, which guarantees that inference results remain the same regardless of batch sizes. However, as its name indicated, this technique is currently limited to eliminating variances from batch sizes. Yet, as discussed above, many other factors, often even harder to control, can also contribute to nondeterminism.

In this work, we aim to achieve *fully* deterministic LLM inference and ensure bit-wise alignment between rollout and training in RL pipelines. We found that tensor parallelism is one of the most critical and practically unresolved sources of nondeterminism, representing the last missing piece toward fully deterministic inference.

To address this challenge, our key idea is to align the accumulation order across GPUs, ensuring that matrix multiplications and inter-GPU reductions follow a consistent arithmetic sequence regardless of TP size. Based on this idea, we propose **Tree-Based Invariant Kernels (TBIK)**, which achieve fully deterministic inference and bit-wise alignment between vLLM and FSDP in RL pipelines.

In summary, our contributions are threefold.:

- Identify tensor-parallelism-induced nondeterminism under varying TP sizes and analyze its impact on benchmark evaluation and RL training.

- Propose Tree-Based Invariant Kernels, which enforce a fixed and consistent computation order for both intra-

GPU matrix multiplications and inter-GPU All-Reduce operations, thereby eliminating TP-induced nondeterminism.

- Enable fully deterministic inference and address probability discrepancies in RL pipelines by integrating TBIK into vLLM and FSDP, resulting in truly on-policy RL training.

## 2. Background

### 2.1. IEEE 754 FP Operations are Non-Associative

A widely known issue associated with IEEE 754 floating point operations is that they are *non-associative*, namely, $(a + b) + c \neq a + (b + c)$ (Yuan et al., 2025a). This means that the order in which numbers are added can affect the final result due to accumulated rounding errors.

Many algorithm and system-level efforts are employed to mitigate this issue. Fused Multiply-Add (FMA) (NVIDIA Corporation, 2025) is used to compute $x*y+z$ with an exact double-length product, followed by an addition with a single rounding. In MatMul and Flash-Attention (Dao et al., 2022) kernels, a common practice is to use FP32 accumulator for reducing intermediate results. Other commonly used operations like Softmax, RMSNorm, and RoPE, are all set to FP32 by default.

### 2.2. Sources of Nondeterminism

In modern serving systems, there are several factors that can change the order of FP operations, thus affecting final results. *(1) Continuous batching* (Yu et al., 2022), which dynamically changes the set of requests in a batch and the

batch size. *(2) Different implementations of operations*, such as the use of Split-K versus Non-Split-K MatMul (NVIDIA Corporation), yield nondeterministic results because Split-K requires accumulating partial sums whose combination order varies based on thread scheduling. *(3) Hyperparameters of operations*, like the block size for MatMul and Flash-attention, also change the specific sequence of accumulation steps within a kernel. *(4) Collective operations in parallel systems*, like All-Reduce, are often nondeterministic, causing the final aggregated value to differ. *(5) Parallel strategies* like TP, which shard workloads across GPUs. *(6) Different GPU architectures*, which may rely on different low-level instruction sets for MatMul, such as `wgmma` on Hopper.

Due to these factors, prior work (Yuan et al., 2025a) shows that even under greedy decoding, inference outputs can diverge significantly across different batch sizes, GPU architectures, and TP configurations.

## 3. Motivation and Challenges

### 3.1. Motivation: Nondeterministism Impact to RL

Nondeterminism can impact many scenarios as mentioned in Section 1. Here, we use the most popular area, LLM-based Reinforcement Learning (RL) as an example to showcase its impact. For the ease of illustration, following (Yao et al., 2025a), we use the REINFORCE algorithm as an example which updates the policy $\pi_\theta$ via

$$\theta \leftarrow \theta + \mu \cdot \underbrace{\mathbb{E}_{a \sim \pi(\theta)}}_{\text{rollout}} \left[ R(a) \cdot \underbrace{\nabla_\theta \log \pi(a, \theta)}_{\text{training}} \right],$$

where $a$ is the token generated by the policy network $\pi_\theta$ and $R(a)$ is the reward. In practice, rollout generation is executed by inference engines such as vLLM and SGLang (Zheng et al., 2024), while model training uses a separate backend (e.g., FSDP). Such a hybrid design update the parameters as follows:

$$\theta \leftarrow \theta + \mu \cdot \mathbb{E}_{a \sim \pi_{\text{rollout}}(\theta)} \left[ R(a) \cdot \nabla_\theta \log \pi_{\text{learner}}(a, \theta) \right].$$

Even if $\pi_{\text{sampler}}$ and $\pi_{\text{learner}}$ share an identical parameters $\theta$, differences in kernel implementations and framework settings (e.g., TP size) can result in large discrepancies in probability calculations as shown in Figure 1, which implicitly converts an on-policy update into an off-policy one (Yao et al., 2025a; Li et al., 2025). This mismatch can destabilize training and negatively affect convergence and policy performance. Note that this problem is much more severe in Mixture-of-Experts (MoE) models (Fedus et al., 2022; Yang et al., 2025) compared to dense models, as a small perturbation in probability may cause the router to

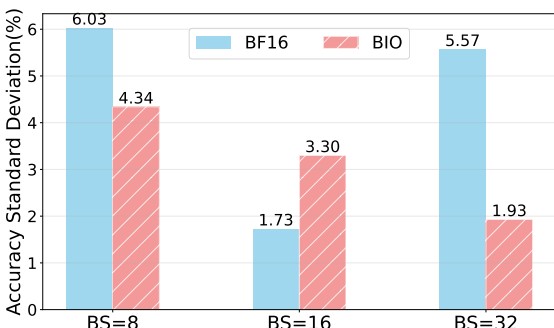

*Figure 2.* Accuracy standard deviation of Qwen3-8B on AIME24 dataset under different TP sizes (TP = 1/2/4/8).

select entirely different experts. An extended discussion of related work on RL mismatch is provided in Appendix A.

### 3.2. Batch-Invariant Operations and System Parallelism

To address this problem, (He & Lab, 2025) introduced *batch-invariant operations* (BIO), including batch-invariant Flex-Attention, RMSNorm, and MatMul kernels, which guarantees that inference results remain deterministic regardless of batch sizes. Specifically, it achieves this by parallelizing the computation along the batch dimension and making each batch element compute independently, which guarantees a fixed reduction order regardless of the batch size. For example, in RMSNorm, each batch element is processed on a single compute core, eliminating inter-core communication for feature-dimension reductions. In MatMul, each core computes the dot products for a 2D tile and performs the full reduction locally, following a non-Split-K strategy. See Appendix B for a schematic illustration.

However, "batch-invariant" is only the first step towards achieving fully deterministic inference. As we explained in Section 2.2, there are many other factors contributing to nondeterminism. The most important ones are those related to parallelisms. To understand the problem, here, we first provide a brief overview of the three most commonly used parallel strategies in serving systems and discuss how BIO behave under each of them.

**Data Parallelism (DP).** Each GPU holds a complete copy of the model but processes a different subset of batch samples, which effectively shards the workload along the batch-size dimension. Changing the degree of DP parallelism is therefore equivalent to changing the batch size, and BIO ensure results to remain consistent.

**Pipeline Parallelism (PP).** Different layers of the model are distributed across multiple GPUs or nodes, **which is the default inter-node parallelization strategy**. The intermediate activations are sequentially passed to the next stage to complete the forward computation. So, changing the PP

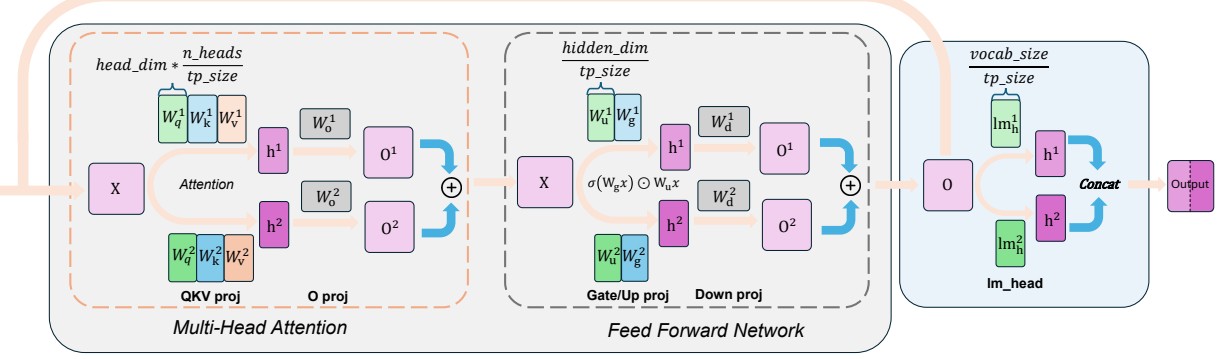

*Figure 3.* Illustration of tensor-parallel weight partitioning in the Transformer model architecture (e.g., Qwen3 Dense) in vLLM. In this configuration, the *QKV proj*, *gate proj*, *up proj*, and *lm head* layers are *column-parallel*, while the *o proj* and *down proj* layers are *row-parallel*. For non-MatMul operations such as RMSNorm and RoPE, parameters and computations are not split across GPUs, and these layers are omitted from the figure for clarity.

degree does not affect determinism.

**Tensor Parallelism (TP).** The weight matrix of each layer is sharded across multiple GPUs, and the complete output of the layer is obtained by aggregating the partial results computed on different GPUs. **TP is the default intra-node parallelization strategy**. As we analyze later in Section 3.3, changing the TP configuration can alter the model's outputs, and batch-invariant kernels are not effective in handling such variations. We evaluate the same prompts from the AIME24 dataset under four TP settings (1/2/4/8) and find that the outputs are all different across these configurations. Moreover, solely changing the TP size leads to over 4% accuracy variation on the AIME24 dataset for Qwen3-8B, as shown in Figure 2.

It is worth noting that TP is widely used in LLM serving systems, and its size is routinely tuned for performance based on different environments. TP size mismatches are particularly unavoidable in RL pipelines, where rollout engines typically use large TP sizes to maximize inference throughput, while training engines often adopt different strategies, such as FSDP with TP=1, to improve memory efficiency.

### 3.3. Why BIO Fail Under Different TP Sizes?

In this section, we first explain the weight-sharding strategy employed in tensor parallelism and then identify the underlying reasons why batch-invariant operations fail to ensure TP invariance.

Figure 3 illustrates how the weights of transformer models are split under tensor parallel in vLLM. In general, TP distributes workloads in two complementary ways: **Column Parallel** and **Row Parallel**. Specifically, the *QKV proj* in self-attention and the *up proj* and *gate proj* in the feed-forward network (FFN) are implemented as **column-**

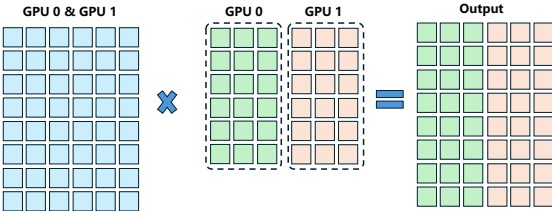

*(a)* Column-parallel matrix multiplication.

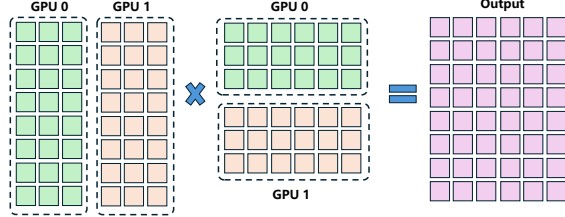

*(b)* Row-parallel matrix multiplication.

*Figure 4.* Illustrations of column-parallel and row-parallel MatMul.

**parallel** operations. In column-parallel mode (Figure 4a), the weight matrix $W \in \mathbb{R}^{K \times N}$ is split along its output dimension into $C$ blocks, where $C$ is the number of GPUs:

$$W_i \in \mathbb{R}^{K \times \frac{N}{C}}, \quad i = 1, \ldots, C,$$

Each GPU computes a partial result

$$O_i = XW_i, \quad O_i \in \mathbb{R}^{M \times \frac{N}{C}},$$

and the output is obtained by stacking all partial results:

$$O = XW = \begin{bmatrix} XW_1 & XW_2 & \cdots & XW_C \end{bmatrix}.$$

In contrast, the *o proj* in attention and the *down proj* in the FFN are **row-parallel**. In row-parallel mode (Figure 4b), the

input matrix $X$ is split along its columns while the weight matrix $W$ is split along its rows, so that each GPU holds a slice

$$X_i \in \mathbb{R}^{M \times \frac{K}{C}}, \quad W_i \in \mathbb{R}^{\frac{K}{C} \times N}, \quad i = 1, \ldots, C,$$

Each GPU computes the partial product $X_i W_i$, and the final output matrix is obtained by summing all partial results across GPUs:

$$XW = \sum_{i=1}^{C} X_i W_i.$$

This design requires a collective `all-reduce` operation to merge the partial results, making the output sensitive to the order of summation.

The rationale behind this design is rooted in how data flows through Transformer layers. The outputs of column-parallel layers (e.g., the *up proj*) serve as inputs to subsequent row-parallel layers (e.g., the *down proj*). Because the column-parallel layers naturally partition their outputs along the feature dimension, these partitions can be directly consumed by the row-parallel layers without additional synchronization. However, the row-parallel linear layer, which serves as the final layer in both the self-attention and FFN modules, must aggregate results from all GPUs through `all-reduce` operations to obtain the complete output of the current module.

**Changing the TP size alters the computation order of the row-parallel layer**. For the column-parallel layer, changing the TP size does not affect the output, since the outputs from each GPU are mutually independent and no accumulation occurs along the shared dimension. However, in the case of a row-parallel layer, both the input $X$ and the weights $W$ are partitioned across GPUs along the $K$ dimension. Each GPU computes a partial result of size $M \times N$, and the layers aggregate all partial results across GPUs through element-wise reductions. Even if the reduction algorithm itself is deterministic, the number of participating devices can alter the computation order for each element. As illustrated in Figure 5, when using the standard cuBLAS GEMM for matrix multiplication, each GPU first performs a local reduction along the $K$ dimension sequentially, after which the partial results are further reduced across GPUs via NCCL along the same dimension. Thus, different TP sizes change the reduction order, which can lead to divergent outputs. Consequently, BIO alone cannot guarantee reproducibility across different TP configurations.

## 4. Tree-Based Invariant Kernels

In this section, we introduce the design and implementation of **Tree-Based Invariant Kernels (TBIK)**, which achieve bit-wise deterministic inference across varying TP sizes.

### 4.1. Overview and Design Principle

To achieve fully deterministic inference across different TP sizes, intra-GPU MatMul and inter-GPU reduction must be co-designed to ensure that the global accumulation order of floating-point operations is invariant. Based on this insight, we propose **TBIK**, which enforces a fixed hierarchical reduction structure spanning both intra-GPU and inter-GPU computation.

The core principle of TBIK is to make the accumulation order independent of the TP size. Specifically, we impose a fixed full binary-tree reduction topology shared by local MatMul and distributed collective operations. Each partial MatMul result corresponds to a leaf node, while internal nodes perform deterministic pairwise accumulation. Because the reduction tree is fixed and does not depend on the number of GPUs, all TP configurations follow identical accumulation paths, thereby guaranteeing deterministic results. A theoretical proof is provided in Appendix C.

### 4.2. Components of TBIK

The overall workflow of TBIK is illustrated in Figure 5, which comprises two stages: (i) intra-GPU reduction within a Tree-Based MatMul kernel, as shown in Figure 11 and (ii) inter-GPU reduction with a custom Tree-Based All-Reduce kernel that follows the same tree topology.

**Intra-GPU Reduction.** We implement the intra-GPU reduction using `Triton` (Tillet et al., 2019), where MatMul is computed in tiles. Specifically, the input matrices are partitioned into tiles along the $K$ dimension, and we denote by $T_K$ the total number of such tiles. Under tensor parallelism with $C$ GPUs, each GPU processes $\frac{T_K}{C}$ tiles.

To ensure a deterministic local accumulation order, we organize the intra-GPU reduction as a fixed binary tree rather than a sequential accumulation. Intermediate partial results are stored at different levels of the tree, whose depth is $L = \log_2 \frac{T_K}{C}$. Each partial MatMul result corresponds to a leaf node, and internal nodes perform pairwise accumulation according to the predefined tree topology.

Because the reduction tree is fixed and independent of the TP size, the accumulation order within each GPU remains identical across all TP configurations. The detailed kernel implementation is described in Appendix D.1.

**Inter-GPU Reduction.** After local accumulation, per-GPU partial results are synchronized using a Tree-Bas all-reduce operation. Crucially, the inter-GPU reduction follows the same binary-tree topology as the intra-GPU reduction, preserving the global accumulation order. The implementation of tree all-reduce is provided in Appendix D.2.

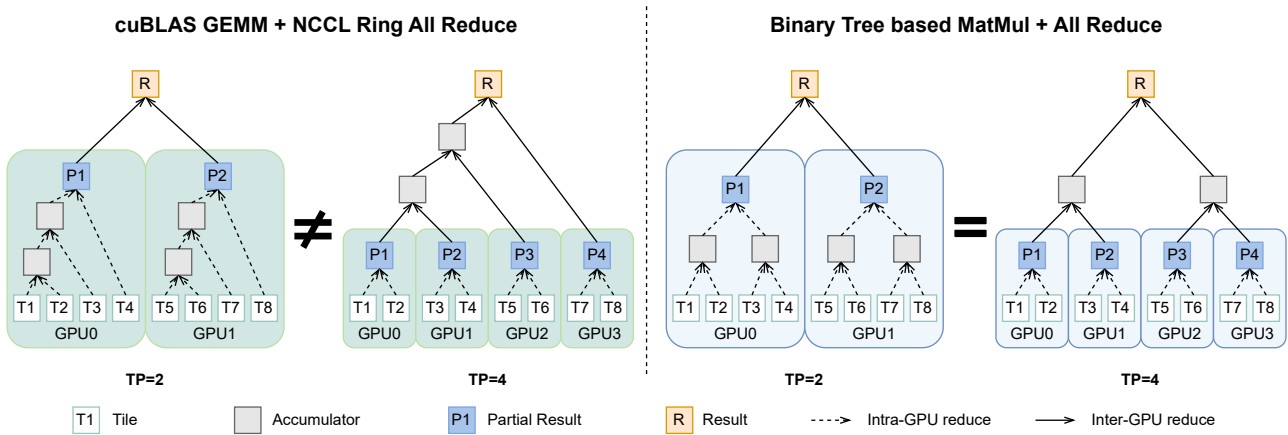

*Figure 5.* Comparison between the cuBLAS GEMM with NCCL Ring Reduce (left) and our Tree-Based Invariant Kernels (right). In the Ring Reduce, inter-GPU communication follows a ring pattern, while intra-GPU reduction is performed sequentially. In contrast, TBIK adopts a hierarchical tree structure for both intra- and inter-GPU reductions, ensuring a consistent reduction order across different TP sizes.

## 5. Evaluation

We designed our experiments to mainly answer three research questions: **RQ1.** Whether our proposed methods can achieve bit-wise identical results across different TP sizes? **RQ2.** What is the overhead introduced by our methods? **RQ3.** Can our methods fundamentally solve the precision mismatch problem in RL training?

### 5.1. Evaluation Metrics

We evaluate the reproducibility of model outputs using two metrics: **Count of Unique Outputs**, and **Maximum Probability Divergence**.

**Count of Unique Outputs:** We count the number of distinct generated token sequences for the same prompt across $K$ runs:

$$U = |\text{unique}(\{y_1, y_2, \ldots, y_K\})|.$$

A value of $U = 1$ indicates outputs are the same across settings.

**Maximum Probability Divergence:** To evaluate bit-wise reproducibility, we compute $\Delta_i$, the maximum divergence of the top-5 predictive probabilities at position $i \in \{1, \ldots, L\}$ across $K$ experimental settings, and report the average divergence over all positions. Bit-wise identical outputs yield a value of zero.

### 5.2. Experiment Setup

We conduct experiments on four models from different families: Qwen3-8B, Qwen3-32B (Yang et al., 2025), Mistral-7B-Instruct-v0.3 (Mistral AI, 2024), and Llama-3.1-8B-Instruct (Meta AI, 2024). The Qwen models are evaluated under the thinking mode with a maximum output length of 8192, while the instruct models are evaluated with a maximum output length of 2048. We evaluate our models on two commonly used benchmarks, AIME24 (Jia, 2024) and AMC23 (AI-MO, 2024), which test numerical reasoning and mathematical problem-solving capabilities. We evaluate the reproducibility of LLMs under random sampling with fixed random seeds and decoding parameters, which better reflects real-world usage scenarios.

Our experiments evaluate three methods: (1) vanilla BF16 inference, (2) inference using only **B**atch-**I**nvariant **O**perations (**BIO**), and (3) inference combining our **T**ree **B**ased **I**nvariant **K**ernels with **B**atch-**I**nvariant **O**perations (**BIO+TBIK**). For **BIO**, we use the implementation from (He & Lab, 2025). For each model–dataset pair, we evaluate under 12 different runtime configurations, representing all combinations of 4 TP sizes (1/2/4/8), and 3 batch sizes (8/16/32), to simulate diverse deployment environments commonly encountered in real-world inference. For the Qwen3-32B model, due to GPU memory limitations, we adopt 9 runtime configurations with 3 TP settings (2/4/8), and 3 BS settings (8/16/32).

We repeat all the above experiments on two different GPU types, i.e., NVIDIA RTX PRO 6000 and NVIDIA L40S, to verify the consistency of results across heterogeneous hardware environments. Experiments are conducted using vLLM (Kwon et al., 2023) as the inference backend. More details about experiment setup, please refer to Appendix E and F.

### 5.3. Reproducibility Evaluation

As shown in Table 1, under vanilla BF16 inference, the average Count of Unique Outputs is close to the number of different runtime configurations, indicating that changing

*Table 1.* **Average Count of Unique Outputs on AIME24 and AMC23.** For each prompt, outputs are generated under 12 runtime configurations (BS=8/16/32; TP=1/2/4/8). Qwen3-32B was evaluated under 9 configurations (BS=8/16/32; TP=2/4/8). "1" indicates identical outputs across all configurations.

| Model | Method | AIME'24 | AMC'23 |
|---|---|---|---|
| Qwen3-8B | BF16 | 12.00 | 10.85 |
| | BIO | 7.87 | 7.78 |
| | BIO+TBIK | **1** | **1** |
| Mistral-7B-Instruct | BF16 | 12.00 | 11.08 |
| | BIO | 7.97 | 7.75 |
| | BIO+TBIK | **1** | **1** |
| Llama-3.1-8B-Instruct | BF16 | 9.60 | 9.85 |
| | BIO | 7.20 | 7.13 |
| | BIO+TBIK | **1** | **1** |
| Qwen3-32B | BF16 | 9.00 | 8.00 |
| | BIO | 6.90 | 6.75 |
| | BIO+TBIK | **1** | **1** |

*Table 2.* **Average Maximum Probability Divergence on AIME24 and AMC23**($\times 10^{-3}$). A larger value indicates greater numerical discrepancies, while 0 means bit-wise identical outputs.

| Model | Method | AIME'24 | AMC'23 |
|---|---|---|---|
| Qwen3-8B | BF16 | 10.01 | 6.79 |
| | BIO | 9.10 | 6.90 |
| | BIO+TBIK | **0** | **0** |
| Mistral-7B-Instruct | BF16 | 17.37 | 19.88 |
| | BIO | 14.10 | 15.87 |
| | BIO+TBIK | **0** | **0** |
| Llama-3.1-8B-Instruct | BF16 | 26.48 | 31.09 |
| | BIO | 27.54 | 21.06 |
| | BIO+TBIK | **0** | **0** |
| Qwen3-32B | BF16 | 8.01 | 9.80 |
| | BIO | 7.79 | 8.97 |
| | BIO+TBIK | **0** | **0** |

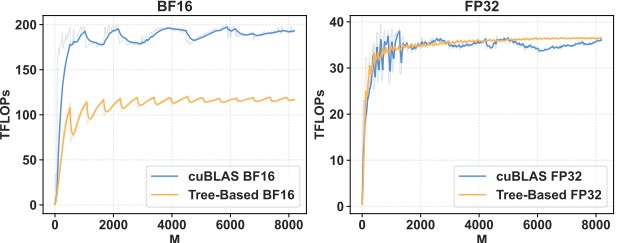

*Figure 6.* Throughput comparison between our Tree-Based MatMul kernels and cuBLAS MatMul kernels on BF16 and FP32 as $M$ varies, with $K = 6144$ and $N = 2048$.

any single system configuration leads to different outputs. When only BIO are applied, the average Count of Unique Outputs decreases, since consistent outputs are obtained across different batch sizes when TP equals 1 or 2, reflecting the "batch invariance" property[1]. However, this approach fails to maintain consistency when $TP > 2$ or when different TP sizes are used. In contrast, our BIO+TBIK setting consistently produces identical inference outputs across all system configurations.

Table 2 reports the average Maximum Probability Divergence on two datasets. Compared with vanilla BF16 inference, BIO slightly reduces probability divergence but still exhibits non-negligible discrepancies. In contrast, BIO+TBIK achieves a strictly zero Maximum Probability Divergence across all experimental settings, demonstrating bit-wise deterministic LLM inference. The above results were obtained on NVIDIA L40s, more experimental results on NVIDIA RTX PRO 6000 can be found in the Appendix G.

### 5.4. Performance Evaluation

In this section, we present the performance evaluation of our Tree-Based MatMul Kernel, as well as the end-to-end latency after integrating TBIK into the SGLang (Zheng et al., 2024) backend.

**Kernel-Level Comparison.** For the kernel throughput analysis shown in Figure 6, we observe that our MatMul kernel runs slower than cuBLAS at small $M$ due to the fixed block-size constraint, which introduces noticeable compu-

tation overhead. As the batch size increases, our kernel achieves 63% of cuBLAS performance, reaching around 120 TFLOPS in BF16 compared to 190 TFLOPS for cuBLAS. This overhead primarily stems from the additional temporary accumulators required for tree-reduction operations, which increase both I/O and arithmetic operations. Under the FP32 setting, the impact of these overheads is mitigated, and both kernels achieve comparable performance. **We emphasize that our current implementation of Tree-Based MatMul kernels primarily serves to demonstrate that TP-Invariant deterministic inference is achievable.** The performance can be further improved by incorporating ad-

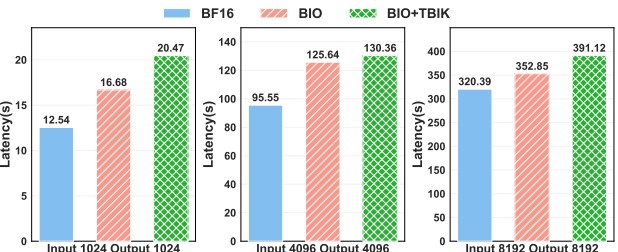

*Figure 7.* End-to-end latency of the Qwen3-8B model on four NVIDIA H20 GPUs with NVLink, evaluated under varying input and output lengths with a batch size of 64.

---

[1]To clarify, BIO ensures consistent results across different batch sizes when running with either TP=1 or TP=2. However, the outputs produced under TP=1 and TP=2 are not identical to each other.

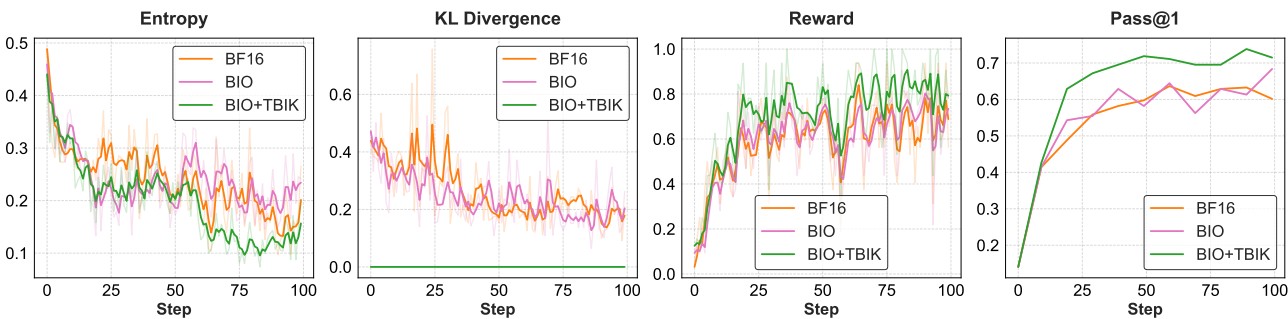

*Figure 8.* RL training(GRPO) metrics on GSM8K using Qwen3-1.7B with four L40S GPUs.

vanced optimizations such as block size tuning, and warp specialization, which have been shown to significantly enhance performance (Yu et al., 2025).

**End-to-End Latency Comparison.** We measured the end-to-end latency of the Qwen3-8B model using four NVIDIA H20 (TP=4) with NVLink across different input and output lengths, with a batch size of 64. For BIO, we use implementation by SGLang. As shown in Figure 7, fully deterministic inference using BIO+TBIK introduces substantial overhead compared with vanilla BF16 inference. The overhead can be clearly attributed to two components: the BIO module alone contributes about $10 \sim 33\%$ overhead relative to BF16, while pure TBIK (the additional cost on top of BIO) contributes another $5 \sim 30\%$ relative to BF16. Taken together, these effects result in a total BIO+TBIK overhead ranging from $22\%$ to $63\%$ compared to the vanilla BF16 baseline. For TBIK, the contributors to this cost are the lower throughput of the unoptimized Tree-Based MatMul compared to cuBLAS BF16, and the sub-optimal deterministic Tree-Based All-Reduce operation we employ. A fine-grained decomposition of TBIK's computational costs can be found in Appendix H. It reveals that the MatMul kernel accounts for $2 \sim 25\%$ of the overhead and the All-Reduce operation incurs a maximum of $10\%$ overhead.

In summary, we show that the performance of deterministic inference is acceptable. Despite the substantial overhead introduced by deterministic inference compared with normal-mode execution, this trade-off is indispensable for scenarios that demand reliable debugging and reproducible results. The performance can be further enhanced by applying advanced optimizations to the Tree-Based MatMul kernel and Tree-Based All-Reduce components of TBIK, in addition to improved integration and more comprehensive optimization for the BIO module within vLLM and SGLang.

### 5.5. Bridging the Probability Gap Between vLLM and FSDP in RL

Here we integrate TBIK into the RL training pipeline to solve the probability mismatch problem between training

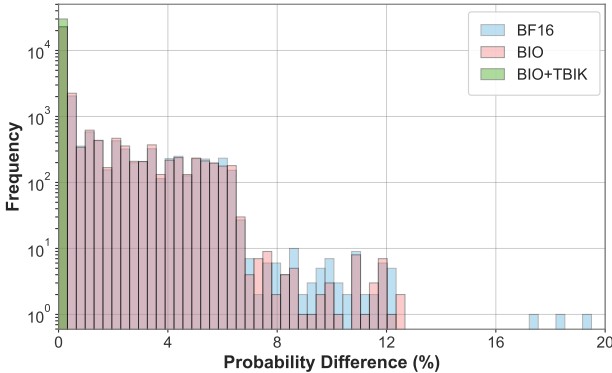

*Figure 9.* Statistics of per-token probability differences between vLLM (TP = 4) and FSDP (TP = 1) on Qwen3-8B with four NVIDIA L40S GPUs. Our method eliminates all probability gaps.

and rollout engines. Specifically, we patch our kernels to both vLLM and FSDP. The implementation details is provided in the Appendix I.1.

As a result, we successfully achieve full determinism across both **vLLM** and **FSDP**. We first examine the per-token probability differences between the two engines on AIME prompts. As shown in Figure 9, BIO slightly reduces the probability discrepancy between the two engines, but a noticeable gap still remains. In contrast, TBIK completely **eliminates the probability gap**, producing bit-wise identical probabilities across all tokens.

We then evaluate our approach on RL training using the GSM8K dataset (Cobbe et al., 2021) with four L40S GPUs, where vLLM performs rollout with a TP size of 4. Detailed training configurations are provided in Appendix I.2. We measure key RL metrics, including **Entropy**, **KL Divergence** between the rollout and training engines, **Reward**, and **Pass@1** rate. As shown in Figure 8, the vanilla BF16 setting exhibits a persistent mismatch between the rollout and training engines, resulting in a nonzero KL divergence throughout training. BIO improves stability and reduces the divergence, but a noticeable gap still remains. In contrast, TBIK achieves zero KL divergence, indicating **bit-wise**

**identical results between rollout and training**, which enables faster convergence, consistently higher rewards, and improved Pass@1 performance. Notably, TBIK surpasses a Pass@1 of 0.6 within the first 20 training steps and ultimately reaches 0.73, compared to 0.68 for BIO and 0.60 for BF16.

## 6. Conclusion and Future Work

We presented TBIK, a framework that achieves deterministic LLM inference across different TP sizes. By enforcing a unified binary-tree reduction order for both intra- and inter-GPU computations, TBIK eliminates nondeterminism introduced by tensor parallelism. When integrated into vLLM and FSDP, TBIK produces bit-wise identical outputs across TP configurations and frameworks. More importantly, by eliminating the probability mismatch between the rollout and training engines, TBIK enables true on-policy RL in practical multi-GPU settings. Our RL experiments show faster convergence and stronger final performance, demonstrating that closing this numerical gap is critical for stable and effective RL training.

Looking forward, a natural extension of this work is to support quantized data types commonly used in efficient LLM(Lin et al., 2024; Frantar et al., 2022; Liu et al., 2024; Yuan et al., 2024). By bringing the same determinism guarantees to quantized settings, we hope to elevate deterministic inference from a "good-to-have" property to a "must-have" requirement for reliable evaluation and on-policy RL training.

## Impact Statement

This paper presents work whose goal is to advance the field of machine learning by improving the determinism and reproducibility of large language model inference and reinforcement learning pipelines. The primary impact of this work is technical, aiming to enhance experimental reliability, fair benchmarking, and stable on-policy reinforcement learning.

We do not foresee immediate negative ethical implications arising directly from this work. More deterministic and reproducible systems may, in fact, support responsible research practices by making results easier to verify and compare. As with most advances in large-scale machine learning systems, the broader societal consequences depend on downstream applications of the models themselves, which are outside the scope of this paper.

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

## A. Related Work on RL mismatch

Beyond our specific kernel-level solution, the community has proposed several effective algorithm-level fixes to stabilize RL training, particularly in the presence of Training-Inference Mismatch. These methods include adopting FP16 as the model dtype to reduce numerical divergence caused by BF16 precision errors(Qi et al., 2025); applying Sequence-level Importance Sampling (SIS) (Li et al., 2025) to compute importance weights over entire generated sequences and use sequence-level reweighting or masked IS to correct distribution shift across trajectories; using Truncated Importance Sampling (TIS)(Yao et al.) to compute token-level importance ratios and truncate them to correct probability discrepancies in a stable way; and utilizing Group Sequence Policy Optimization (GSPO)(Zheng et al., 2025), which performs optimization at the sequence level rather than token level, lowering variance and stabilizing training. Nevertheless, these methods only partially alleviate the mismatch and do not realize bitwise identical true on-policy reinforcement learning.

## B. Batch Invariant Operations

Batch Invariant Operations implement kernels to make each sample follows a fixed computation path independent of the batch size. For RMSNorm, each token is handled independently, and the reduction over the hidden dimension is performed with a fixed accumulation order. For MatMul, each input row is computed independently with a fixed tiling strategy and a fixed reduction order along the $K$ dimension. Therefore, changing the batch size only changes how many rows are launched, but does not change the computation order of any individual row. Figure 10 illustrates the BIO implementations for RMSNorm and MatMul.

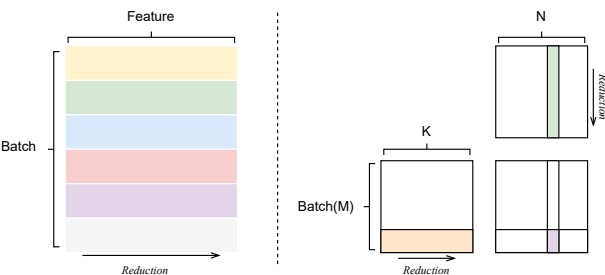

*Figure 10.* Implementations in Batch Invariant Operations. Left: RMSNorm. Right: MatMul.

## C. Theoretical Proof

Here, we provide a theoretical proof of the correctness of TBIK.

**Operator definition.** Define $T(\cdot)$ on sequences by recursion on the full binary-tree topology:

$$T(k_1,\ldots,k_{2t}) = \begin{cases} k_1, & t=0, \\ k_1 \oplus k_2, & t=1, \\ T\big(T(k_1,\ldots,k_{2t-1}), \ T(k_{2t-1+1},\ldots,k_{2t})\big), & t>1. \end{cases}$$

That is, $T(\cdot)$ performs pairwise reductions following the parent–child structure of $\mathcal{T}^*$, where each internal node applies the operator $\oplus$ to the outputs of its two child subtrees.

**Theorem 1.** *Let $N$ be the total number of tiles and $C$ be the TP size, where $C$ is a power of two. If the $N$ tiles are evenly partitioned across the $C$ GPUs, then under the operator $T(\cdot)$ defined above, the hierarchical reduction order is fixed regardless of TP sizes.*

*Proof sketch.* Define $M = N/C$ as the number of tiles per GPU, and assign each GPU $d = 1,\ldots,C$ the contiguous tiles

$$\mathcal{L}_d = \{k_{(d-1)M+1},\ldots,k_{dM}\}.$$

We prove that

$$T(k_1,\ldots,k_N) = T\big(T(\mathcal{L}_1),\ldots,T(\mathcal{L}_C)\big),$$

where $T(\cdot)$ is the binary-tree operator defined above.

**Base case ($C=1$).** When there is only one GPU, $\mathcal{L}_1$ contains all tiles, so

$$T(k_1,\ldots,k_N) = T(\mathcal{L}_1),$$

and the statement trivially holds.

**Inductive step.** Assume the equality holds for $C/2$ GPUs. For $C$ GPUs, split the GPUs into the first half (blocks $\mathcal{L}_1,\ldots,\mathcal{L}_{C/2}$) and the second half (blocks $\mathcal{L}_{C/2+1},\ldots,\mathcal{L}_C$). By the induction hypothesis, the reductions within each half satisfy

$$T(k_1,\ldots,k_{N/2}) = T(T(\mathcal{L}_1),\ldots,T(\mathcal{L}_{C/2}))$$
$$T(k_{N/2+1},\ldots,k_N) = T(T(\mathcal{L}_{C/2+1}),\ldots,T(\mathcal{L}_C)).$$

The global reduction applies $T$ to these two halves:

$$T(k_1,\ldots,k_N) = T\big(T(k_1,\ldots,k_{N/2}),T(k_{N/2+1},\ldots,k_N)\big),$$

which by the induction hypothesis equals

$$T\big(T(\mathcal{L}_1),\ldots,T(\mathcal{L}_C)\big).$$

Hence, by recursion, the hierarchical reduction executes the exact same sequence of pairwise $\oplus$ operations as the global reduction, so the result is bitwise-identical. $\square$

# D. Implementation Details of TBIK

## D.1. Tree-Based MatMul Kernel

To realize a tree-structured accumulation with a fixed reduction order, intermediate partial results must be temporarily stored. In TBIK, each GPU is responsible for processing $\frac{T_K}{C}$ tiles along the reduction dimension, where $T_K$ is the total number of tiles and $C$ denotes the number of GPUs. We show that at least $L$ accumulators are required, where

$$L = \log_2 \frac{T_K}{C}$$

corresponds to the depth of the binary reduction tree. Accordingly, we allocate an accumulator buffer $S$ of shape $[L, \texttt{Block}_M, \texttt{Block}_N]$ for computing each output tile.

During computation, tiles of shape $[\texttt{Block}_M, \texttt{Block}_K]$ from $A$ and $[\texttt{Block}_K, \texttt{Block}_N]$ from $B$ are loaded sequentially. Each partial product is first accumulated into the level-0 buffer of $S$. To track the reduction state, we maintain a counter tensor $\texttt{Count}$ of length $L$, where $\texttt{Count}[l]$ records the number of partial sums currently stored at level $l$. When $\texttt{Count}[l]$ reaches two, a *carry-over* operation is triggered: the two partial sums at level $l$ are reduced and propagated to level $l + 1$ via

$$S[l + 1] = S[l + 1] + S[l],$$

after which both the value and the counter at level $l$ are cleared. This hierarchical reduction proceeds until all tiles along the $K$-dimension are processed, and the final output is stored in $S[L]$.

When $T_K$ is not a power of two, as in the $\texttt{down\_proj}$ layer of Qwen3-1.7B with $K = 6144$ and power-of-two $\texttt{Block}_K$, we introduce an adaptive parameter $K_{\text{first}}$ to control how many tiles are accumulated before the first carry-over. Although this relaxes the strict binary pattern at the first reduction level, the overall accumulation order remains invariant as long as

$$\frac{T_K}{C \times K_{\text{first}}} \geq TP_{\max}.$$

Figure 11 illustrates the overall workflow of the Tree-Based MatMul kernel in TBIK, and Algorithm 1 details its implementation.

## D.2. Tree-Based All-Reduce

We do not directly adopt NCCL's built-in tree all-reduce algorithm because NCCL only allows users to specify a tree topology **across nodes**, while **within each node** it still defaults to a chain-based reduction. In typical deployment scenarios, tensor parallelism is applied to multiple GPUs within the same node. Therefore, to ensure that reductions

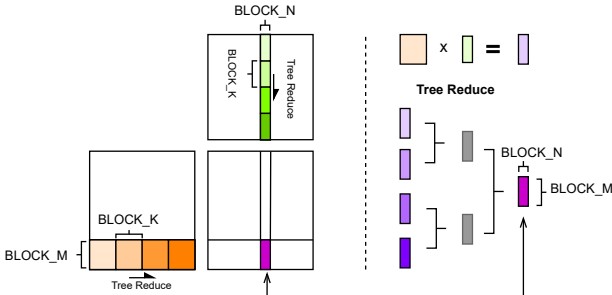

*Figure 11.* Illustration of our Tree-Based MatMul kernel.

within a node also strictly follow a tree-structured topology, we implement a customized tree all-reduce algorithm as shown in Algorithm 2.

# E. Supplementary Experiment Setup Details

## E.1. Decoding Parameters for Random Sampling

The decoding parameters are set to (temperature = 0.6, top-p = 0.95, top-k = 20) for reasoning models, and (temperature = 0.7, top-p = 0.8, top-k = 20) for non-reasoning models following the official best practice provided in the Qwen3 Techinical Report (Yang et al., 2025).

## E.2. Deterministic Inference in vLLM

To enable deterministic inference with vLLM, all our experiments are conducted in eager mode, under which CUDA Graphs are disabled, and prefix caching is turned off. We fix the random seed to 42.

However, the vLLM v1 engine enables chunked prefill by default and does not allow users to disable it (vLLM Project, 2025). This chunking strategy potentially conflicts with the requirements for deterministic inference (LMSYS Organization, 2025). To efficiently handle long prompts under limited GPU memory, vLLM divides the prefill process of a sequence into multiple chunks. This chunked execution alters the computation and scheduling order—even when the underlying computation kernels are deterministic. Moreover, this process is invisible to users and cannot be explicitly controlled.

We observed that these implicit optimizations can indeed affect model outputs in practice. When the model size or batch size are large, GPU memory is constrained. Under these conditions, the inference process is not run-to-run deterministic, and batch-invariant operations (BIO) cannot fully ensure reproducible outputs when the batch size changes, even at TP=1. To eliminate the impact of such serving-system scheduling behaviors on our method evaluation, we set $\texttt{max\_num\_seqs=1}$ for the Qwen3-32B experiments on NVIDIA L40S with TP=2.

**Algorithm 1** Tree-Based MatMul Kernel

**Require:** Matrices $A \in \mathbb{R}^{M \times K}$, $B \in \mathbb{R}^{K \times N}$, block sizes $B_M, B_N, B_K$, total tiles $T = K/B_K$, first level tile count $K_{first}$, reduction depth $L = \log_2(T/K_{first}) + 1$

**Ensure:** Output matrix $C \in \mathbb{R}^{M \times N}$

  **for all** blocks $(m, n)$ in grid $(\lceil M/B_M \rceil, \lceil N/B_N \rceil)$ **in parallel do**

    Initialize local accumulator: $acc[B_M][B_N] \leftarrow 0$

    Initialize scratch buffers: $S[L][B_M][B_N] \leftarrow 0$

    Initialize counters: $Count[L] \leftarrow 0$

    **for** $t = 0$ to $T - 1$ **do**

      Load tile $A_t = A[m:m + B_M, tB_K : (t+1)B_K]$

      Load tile $B_t = B[tB_K : (t+1)B_K, n : n + B_N]$

      $acc \leftarrow acc + A_t B_t$

      $level \leftarrow 0$

      **while** $level < L$ **do**

        **if** $(level = 0$ **and** $Count[level] + 1 = K_{\text{first}})$ **or** $(level > 0$ **and** $Count[level] + 1 = 2)$ **then**

          $acc \leftarrow acc + S[level]$

          $S[level] \leftarrow 0$

          $Count[level] \leftarrow 0$

          $level \leftarrow level + 1$

        **else**

          $S[level] \leftarrow acc$

          $Count[level] \leftarrow Count[level] + 1$

          **break**

        **end if**

      **end while**

    **end for**

    $C[m : m + B_M, n : n + B_N] \leftarrow acc$

  **end for**

---

**Algorithm 2** Tree-Based All-Reduce

**Require:** Local tensor $x$ on rank $r$, world size $W = 2^q$

**Ensure:** All-reduced tensor $x$

  $buf \leftarrow x$

  // Reduce phase

  **for** $\ell = 0$ to $q - 1$ **do**

    $s \leftarrow 2^\ell$

    **if** $r \bmod 2s = 0$ **then**

      $buf \leftarrow buf + \text{RECV}(r + s)$

    **else if** $r \bmod 2s = s$ **then**

      $\text{SEND}(buf, r - s)$

      **break**

    **end if**

  **end for**

  // Broadcast phase

  **for** $\ell = q - 1$ down to $0$ **do**

    $s \leftarrow 2^\ell$

    **if** $r \bmod 2s = 0$ **then**

      $\text{SEND}(buf, r + s)$

    **else if** $r \bmod 2s = s$ **then**

      $buf \leftarrow \text{RECV}(r - s)$

    **end if**

  **end for**

  **return** $buf$

---

# F. TBIK Block Size Selection

In this section, we present the block sizes used in TBIK, namely $Block_M$, $Block_N$, and $Block_K$. To achieve optimal throughput, we employ different configurations for different data types, as summarized in Table 5. The block sizes are determined via a grid search over the values $[16, 32, 64, 128, 256]$ on NVIDIA L40S GPUs, and for each data type we select the configuration that yields the highest throughput.

# G. Supplementary Reproducibility Evaluation Results

## G.1. Average Count of Unique Outputs on NVIDIA RTX PRO 6000 GPU

As shown in Tables 3, vanilla BF16 inference exhibits high nondeterminism under varying system configurations. The BIO can partially mitigate this issue by reducing the average number of unique outputs, as consistent results are

*Table 3.* **Average Count of Unique Outputs on AIME24 and AMC23.** For each prompt, outputs are generated under 12 runtime configurations (BS=8/16/32; TP=1/2/4/8). Qwen3-32B was evaluated under 9 configurations (BS=8/16/32; TP=2/4/8). "1" indicates identical outputs across all configurations.

| Model | Method | AIME'24 | AMC'23 |
|---|---|---|---|
| Qwen3-8B | BF16 | 12.00 | 11.20 |
| | BIO | 7.87 | 7.53 |
| | BIO+TBIK | **1** | **1** |
| Mistral-7B-Instruct | BF16 | 11.97 | 11.13 |
| | BIO | 7.97 | 7.58 |
| | BIO+TBIK | **1** | **1** |
| Llama-3.1-8B-Instruct | BF16 | 9.37 | 9.55 |
| | BIO | 7.00 | 6.80 |
| | BIO+TBIK | **1** | **1** |
| Qwen3-32B | BF16 | 9.00 | 8.20 |
| | BIO | 6.87 | 6.73 |
| | BIO+TBIK | **1** | **1** |

achieved when the TP size is 1 or 2 with varying batch sizes. When combining BIO with TBIK, the average number of unique outputs across all models on both the AIME24 and AMC23 datasets drops to one, indicating fully deterministic outputs under all tested configurations. This observation is consistent with our experimental findings on NVIDIA L40S GPUs.

## G.2. Average Maximum Probability Divergence on NVIDIA RTX PRO 6000 GPU

*Table 4.* **Average Maximum Probability Divergence on AIME24 and AMC23**($\times 10^{-3}$). A larger value indicates greater numerical discrepancies, while 0 means the predicted token probability distributions are bitwise identical across all evaluated runtime configurations.

| Model | Method | AIME'24 | AMC'23 |
|---|---|---|---|
| Qwen3-8B | BF16 | 9.72 | 7.63 |
| | BIO | 8.35 | 7.14 |
| | BIO+TBIK | **0** | **0** |
| Mistral-7B-Instruct | BF16 | 15.58 | 19.48 |
| | BIO | 14.85 | 17.44 |
| | BIO+TBIK | **0** | **0** |
| Llama-3.1-8B-Instruct | BF16 | 31.84 | 27.04 |
| | BIO | 31.65 | 25.95 |
| | BIO+TBIK | **0** | **0** |
| Qwen3-32B | BF16 | 9.84 | 9.34 |
| | BIO | 8.32 | 8.57 |
| | BIO+TBIK | **0** | **0** |

Table 4 present the average Maximum Probability Divergence on two datasets. Compared with vanilla BF16 inference, BIO slightly reduces the fluctuation of token predictive probabilities due to its parallel strategy and control over reduction operations. However, it still fails to handle variations caused by changes in TP. In contrast, BIO+TBIK achieves a strictly zero Maximum Probability Divergence across all experimental settings, demonstrating bit-wise deterministic LLM inference. This observation is consistent with our experimental findings on NVIDIA L40S GPUs.

## H. Supplementary Performance Evaluation Results

### H.1. Fine-grained Latency Breakdown of TBIK

To better understand the contributions of the two core components in TBIK—the Tree-Based MatMul kernel and the Tree-Based All-Reduce operation—to the overall end-to-end performance, we evaluate five configurations: (1) vanilla BF16 inference, (2) BIO enabled, (3) BIO with Tree-Based All-Reduce only (without Tree-Based MatMul), and (4) BIO with Tree-Based MatMul only (without Tree-Based

All-Reduce), (5) BIO with both components enabled.

Figure 12 presents the latency breakdown across different input/output lengths. Although the Tree-Based MatMul exhibits lower throughput compared with the cuBLAS BF16 implementation, it introduces only 2–25% overhead. This limited impact arises because the row-split linear layer accounts for only a small fraction of the model's total computation, and therefore its inefficiency has minimal effect on end-to-end latency.

In contrast, the Tree-Based All-Reduce operation incurs a maximum overhead of approximately 10%. This overhead is evaluated on GPUs equipped with NVLink and is mainly due to the current lack of low-level kernel optimizations in our custom implementation, rather than inherent limitations of the tree-based design itself.

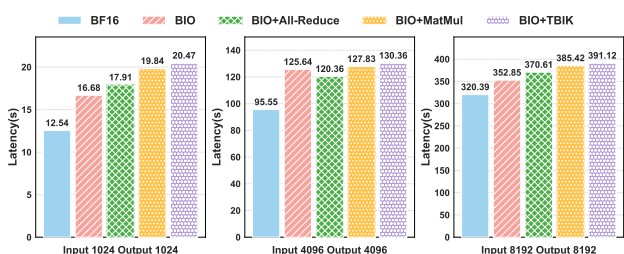

*Figure 12.* Fine-grained end-to-end latency breakdown of the Qwen3-8B model on four NVIDIA H20 GPUs under varying input/output lengths with a batch size of 64.

*Table 5.* Block size setting for TBIK.

| Dtype | BLOCK$_M$ | BLOCK$_K$ | BLOCK$_N$ |
|---|---|---|---|
| BF16 | 64 | 256 | 128 |
| FP16 | 64 | 256 | 128 |
| FP32 | 32 | 128 | 64 |

## I. Implementation and Training Details for On-Policy RL

### I.1. Framework Modifications for vLLM and FSDP

Below we introduce the implementation details besides our kernels to make sure they are aligned. To eliminate these discrepancies, we introduce the following framework-level modifications for both vLLM and FSDP:

1. **Linear Layers.** We replace standard linear layers with customized kernels to ensure both batch and TP invariance. Specifically, we employ BIO for column-parallelized layers in vLLM (`qkv_proj`, `up_proj`, `gate_proj`, and `lm_head`), and our Tree-Based MatMul kernel for row-parallelized layers (`down_proj` and `o_proj`).

2. **Attention.** To ensure determinism, we fix the tile size in TritonAttention to enable batch invariance. Additionally, we disable chunked prefill in vLLM and enforce both the prefill and decode phases to use the same attention kernel. This adjustment is necessary because FSDP training only involves the prefill phase; therefore, we disable these decoding-side optimizations in vLLM. The same modified attention backend is also applied to FSDP.

3. **Other Kernels.** For other kernels, including RMSNorm, RoPE embedding, and SiLU activation, we used the same kernel implementations as in vLLM for FSDP to ensure consistency.

## I.2. RL Training Details

We evaluate TBIK in an online RL setting using GRPO on the GSM8K dataset. All experiments are conducted on four NVIDIA L40s GPUs, where vLLM runs with TP size 4 and FSDP runs with TP size 1.

We sample 256 GSM8K questions for training and 64 questions for evaluation. The maximum number of new tokens is set to 512. We use a group size of 8, a rollout batch size of 64, and a training batch size of 32. The learning rate is fixed at $1 \times 10^{-5}$ throughout training.

