# OpenReview forum: "Deterministic Inference across Tensor Parallel Sizes That Eliminates Training-Inference Mismatch"
_ICML.cc/2026/Conference — ICML 2026 regular_

### Official Review · Reviewer_ygHt · 2026-03-09

**Soundness:** 3
**Presentation:** 3
**Significance:** 2
**Originality:** 2
**Overall Recommendation:** 3
**Confidence:** 4

**Summary:**

This paper investigates deterministic LLM inference in the context of tensor parallelism. The authors identify that a primary source of non-determinism is that varying TP sizes alter the reduction order of floating-point operations, leading to divergent outputs. To address this, the paper introduces a custom GEMM and communication kernel that enforces a consistent binary-tree-based reduction structure. The evaluation demonstrates that the proposed method achieves bit-wise identical results across different TP and batch-size configurations. However, the current kernel performance significantly lags behind cuBLAS, particularly when using BF16 precision.

**Compliance With Llm Reviewing Policy:**

Affirmed.

**Final Justification:**

The rebuttal response did not fully address my concern.

**Key Questions For Authors:**

1. The proposed GEMM kernel achieves only about 60% of cuBLAS performance. Given that the kernel design is one of the paper’s main contributions, this efficiency gap appears significant. In addition, the reported performance results are based on RTX 6000 and L40S GPUs. Do the authors have performance data on more advanced GPUs (e.g., H100)?

2. The theoretical proof and the core algorithm rely on the TP size being a power of two. Does the proposed method support TP sizes that are not powers of two? If not, could the authors discuss whether the method can be extended to such settings, and what challenges would arise?

3. If both the rollout and training systems were set to the same TP configuration (e.g., both using TP=8 and using the same standard kernels), would the precision mismatch be eliminated?

4. In Section 5.2, the paper states that experiments were conducted on RTX 6000 and L40S GPUs. However, Figure 7 reports end-to-end latency using NVIDIA H20 GPUs. Could the authors clarify why different hardware was selected for the end-to-end latency tests?

5. How does the proposed framework interact with other common parallel strategies, such as Sequence Parallelism [1]?

6. Could the authors clarify which specific RL algorithm was used in the experiment shown in Figure 8? It would also strengthen the paper to evaluate on additional RL algorithm, such as GRPO or GSPO.

**Reference:**

[1] Korthikanti, Vijay Anand, et al. "Reducing activation recomputation in large transformer models." Proceedings of Machine Learning and Systems 5 (2023): 341-353.

**Limitations:**

Yes

**Strengths And Weaknesses:**

**Strength:**
* The paper addresses an important and timely systems problem.
* The paper is well organized, and the motivation for the proposed method is clear.
* The proposed method is simple yet effective.

**Weakness:**
* The efficiency of the implemented kernel could be improved further.
* The evaluation is limited in terms of scale, parallelism configurations, and hardware coverage.
* The theoretical proof and the core algorithm rely on the TP size being a power of two.

---

> ### Author Rebuttal · Authors · 2026-03-31
>
> ### [W1&Q1]`Kernel can be further optimized.`--We want to clarify the suboptimal-performance kernels are only applied to two layers, which is provably sufficient to guarantee bitwise-identical results as verified by both theoretical analysis and empirical evaluation. The end-to-end overhead is acceptable.
> We thank the reviewer for this point and want to explain the overhead is acceptable. **TBIK is only applied to two layers per transformer block.** TP-induced non-determinism originates only from the row-parallel linear layers: `o_proj` in attention and `down_proj` in FFN. All other layers run at full cuBLAS speed. **The practical overhead falls at the lower end of this range.** As shown in Figure11, overhead decreases with sequence length. The dominant use case, RL training of reasoning LLMs, operates in the long-sequence regime (>8K tokens), where overhead is ~10%. We note that optimal kernel performance typically requires architecture-specific engineering, which is an orthogonal effort.
> ### [W2]`The evaluation is limited in terms of scale, parallelism config and hardware.`--We want to clarify that our evaluation already covers a meaningful range.
> We thank the reviewer for this feedback and want to clarify our evaluation is sufficient to validate effectiveness of TBIK. **Scale:** Section5.3 covers 8B and 32B models, and Section5.5 includes a 1.7B model. TBIK operates at kernel level, which is fundamentally independent of model size. **Parallelism configurations:** We evaluate TP sizes of 1/2/4/8, covering the vast majority of TP sizes used in practice. TBIK is orthogonal to other parallelism strategies, refer to Q5. **Hardware coverage:** Our experiments span three GPU platforms across three architectures: L40s, RTX Pro 6000 and H20. This diversity validates TBIK's portability across GPU generations.
> ### [W3&Q2]`TP size must be a power of two?`--TP size is almost always a power of 2 in practice.
> We thank the reviewer for this question. **In practice, TP sizes are almost always powers of two** due to 2 independent constraints: *1: TP is limited to intra-node communication.* TP requires an all-reduce at every layer, making it extremely bandwidth-sensitive. It is only used within nodes, and standard GPU servers have 2/4/8 GPUs per node, limiting practical TP sizes to 1/2/4/8. *2: TP must evenly divide the number of attention heads.* TP shards Q/K/V projections by splitting attention heads, requiring `num_heads % TP_size == 0`. With num_kv_heads typically being 4/8(Llama-3, Qwen2.5, Mistral), valid TP sizes are further restricted to small powers of two.
> ### [Q3]`Can same TP configuration solve the issue?`--In theory, yes, but in practice it's not feasible.
> We thank the reviewer for the question. While using identical TP size theoretically eliminates the mismatch, **TP sizes are often different in practice.** The inference side prefers a larger TP size to minimize per-token decoding latency, while the training side favors a smaller TP size to reduce the frequent all-reduce communication overhead on large batches. Forcing the same TP size would mean at least one side operates suboptimally, leading to training inefficiency. **TBIK's core contribution is precisely to decouple TP size from numerical consistency.**
> ### [Q4]`Why report end-to-end latency on H20 GPUs?`--NVLink-enabled hardware is required.
> We thank the reviewer for pointing this out. Our primary development and verification were conducted on L40s and RTX Pro 6000 workstations, which use PCIe interconnects and lack NVLink. In TP, every transformer layer requires an all-reduce operation. Without NVLink, these all-reduce operations become the dominant bottleneck, making it impossible to meaningfully measure the kernel-level overhead differences between TBIK and standard cuBLAS.
> ### [Q5]`How can TBIK interact with other parallel strategies?`--TBIK is compatible with others.
> We thank the reviewer for the question. TBIK works well with other parallel strategies, because **the nondeterminism introduced by TP is not at the same level as that from others.** TP shards individual matrix multiplications across multiple GPUs and uses an all-reduce to obtain the final result, meaning a single token's hidden states are computed collectively across GPUs. In contrast, other parallel strategies do not involve this feature. For example, Sequence Parallelism partitions input data along the token dimension without sharding model parameters. This also applies to PP, DP, CP. Token-level nondeterminism has been addressed by batch-invariant kernels.
> ### [Q6]`What RL algorithm is used in Figure8? Better run more RL algorithm.`--We use GRPO in Figure8.
> The experiments follow the setup of vLLM[1], with the key difference that we set TP size = 4 for inference engine. Much existing work has demonstrated trust-region-based RL algorithms benefit from on-policy data[2].
>
> Reference:
> [1]https://vllm.ai/blog/bitwise-consistent-train-inference
> [2]https://arxiv.org/abs/2510.26788

---

> > ### Author Rebuttal · Reviewer_ygHt · 2026-04-03
> >
> > Thank you for the response.
> >
> > I am still not fully convinced by the claim that trainer and sampler commonly use different TP sizes. In practical, at least on Hopper GPUs, my understanding is that when the model scale is sufficiently large, TP=8 is often a common configuration for both training and inference.

---

> > > ### Author Response · Authors · 2026-04-03
> > >
> > > We thank the reviewer for the response and here provide further evidence to address the concerns. **The TP sizes used during training and inference differ in most mainstream LLMs, such as DeepSeek-V3[1] and Kimi K2.5[2]**. We provide a detailed analysis below.
> > >
> > > **On the training side, the workload is compute-bound.** Training prefers small TP sizes (or no TP at all) because TP requires an all-reduce at every transformer layer, introducing substantial communication volume. Larger TP sizes also reduce per-GPU weight shards, lowering GPU compute utilization and making the compute-communication overlap harder to achieve. This is why many prominent models are trained with TP=1 in practice: DeepSeek-V3 [1] (Section 3.2) and Kimi K2.5 [2] (Appendix C) both explicitly document their training parallel strategies with TP=1.
> > >
> > > **On the inference side, the primary bottleneck is autoregressive decoding, which is memory-bandwidth bound.** Unlike training, decoding involves KV cache and processes only one token per step, reducing per-step computation by an order of magnitude. TP is well-suited for inference because it shards both model weights and KV cache across GPUs, directly reducing per-token decoding latency. DeepSeek-V3 [1] (Section 3.4) documents using TP=4 for inference, and vLLM provides detailed usage guides [3] recommending TP for multi-GPU inference deployment. That said, the optimal inference TP size also depends on model architecture and system configuration. For instance, DeepSeek-V3.2 [4] uses MLA with a single KV head, in which case TP requires replicating the KV cache rather than sharding it, reducing the benefit of large TP.
> > >
> > > These real-world examples demonstrate that TP size mismatch between training and inference is not a hypothetical concern but a practical reality in production LLM systems. TBIK addresses this by decoupling numerical consistency from TP size, allowing both sides to independently choose their optimal parallel strategies.
> > >
> > > Reference:
> > >
> > > [1] https://arxiv.org/abs/2412.19437
> > >
> > > [2] https://arxiv.org/abs/2602.02276
> > >
> > > [3] https://docs.vllm.ai/projects/recipes/en/latest/DeepSeek/DeepSeek-V3.html
> > >
> > > [4] https://docs.vllm.ai/projects/recipes/en/latest/DeepSeek/DeepSeek-V3_2.html#installing-vllm

---

### Official Review · Reviewer_sGip · 2026-03-12

**Soundness:** 3
**Presentation:** 3
**Significance:** 3
**Originality:** 3
**Overall Recommendation:** 5
**Confidence:** 3

**Summary:**

This paper points out TP-induced nondeterminism in LLM inference where identical inputs can yield different outputs when TP size varies, and proposes Tree-Based Invariant Kernels (TBIK) that achieves bit-wise deterministic inference regardless of TP size.

**Compliance With Llm Reviewing Policy:**

Affirmed.

**Final Justification:**

See the Rebuttal Acknowledgement.

**Key Questions For Authors:**

1. Does this issue happen in heterogeneous parallelism during pre-training where across DDP some replicas do FSDP-only and some incorporate TP? It would be interesting to see if incorporating TBIK in this setting would lead to any performance improvements.

2. See Weakenesses.

**Limitations:**

yes

**Strengths And Weaknesses:**

### Strengths

1. The problem of deterministic inference is important and well-motivated, as there is a common TP/FSDP difference in training/rollout.

2. The proposed method is clear and easy to follow, and the paper is well written.

3. The deterministic results are very strong. Especially as shown in Tables 2 and 3, TBIK leads to identical output across all configurations

---

### Weaknesses

1. No end-to-end RL experiment is provided. This experiment would clarify the significance of determinism and how much this improves downstream RL training stability and convergence.

2. The proposed method leads to considerable slowdowns, which is a practical concern. The authors point out current implementation is not optimal and can be improved by advanced optimization techniques. However, it is still unclear to me how the current all-gather in Algorithm 2 can be avoided for an efficient deterministic all-reduce.

3. (minor) There are quite a few typos and malformed citations (e.g. tse Huang, Team, Q etc.)

---

> ### Author Rebuttal · Authors · 2026-03-31
>
> ### [W1] `No end-to-end RL experiment is provided.`--We present end-to-end RL experiment in Figure8, and we provide supplemental results.
>
> Thank you for the valuable feedback. In Figure 8, we report the end-to-end KL divergence between vLLM and FSDP, as well as the reward across different settings. Here, we further report pass@1 across different training steps under the **BF16**, **BIO**, and **BIO+TBIK** settings. **We observe that BIO+TBIK, which achieves true on-policy training, exhibits superior performance and faster convergence compared to the other two baselines.**
>
> For all experiments, we train GRPO with a max sequence length of 512, sampling 256 GSM8K questions for training and 64 for testing. The group size is set to 8, with a rollout batch size of 64 and a training batch size of 32. We use a learning rate of 1e-5.
>
> | Step | BF16 | BIO | BIO+TBIK |
> |---|---|---|---|
> | 0 | 0.1406 | 0.1406 | 0.1406 |
> | 10 | 0.3438 | 0.5312 | 0.6367 |
> | 20 | 0.4023 | 0.6602 | 0.6797 |
> | 30 | 0.5352 | 0.6367 | 0.6875 |
> | 40 | 0.6367 | 0.6875 | 0.6758 |
> | 50 | 0.6680 | 0.6367 | 0.7188 |
> | 60 | 0.7031 | 0.6641 | 0.7383 |
> | 70 | 0.6758 | 0.6523 | 0.7031 |
> | 80 | 0.6641 | 0.6211 | 0.7031 |
> | 90 | 0.6836 | 0.6758 | 0.7070 |
> | 100 | 0.6523 | 0.6758 | 0.7500 |
>
> ### [W2] `How to avoid all-gather operation to improve the efficiency of deterministic all-reduce.`--We use custom cuda kernel instead of NCCL All-Gather.
> Thank you for this question. We sincerely apologize for the confusion caused by not properly updating the information. We have already replaced the all-gather-based implementation with a significantly more efficient custom CUDA kernel. The original implementation used `torch.distributed.all_gather` to collect full data replicas on each GPU, followed by a local tree-structured reduction in Python, which incurred O(N×D) communication and memory overhead plus multiple sequential kernel launches. Our current implementation (`CustomTreeAllreduce`) eliminates the all-gather entirely by using CUDA IPC to enable direct GPU-to-GPU memory access over NVLink. The tree-structured reduction is performed inside a single fused CUDA kernel: each thread directly loads data from remote GPUs through NVLink using 128-bit vectorized reads, performs the pairwise additions in compile-time-unrolled binary tree order, and writes the result to local memory, all within one kernel launch. Inter-GPU synchronization uses lightweight flag-based barriers with release-acquire semantics, avoiding the overhead of NCCL's collective communication stack. We will clarify this implementation upgrade in the revised paper.
> ### [W3] `There are a few typos and malformed citations.`--Thanks for pointing out and we will fix them.
>
> ### [Q1] `Does TP-size mismatch happen during pretraining?`--Interesting idea! Here we provide some discussions.
> Thank you for this insightful suggestion. Heterogeneous parallelism during pre-training does indeed introduce the same numerical inconsistency that TBIK addresses. Replicas with different TP sizes would produce numerically different activations and gradients for the same input data, and when averaged via DDP's all-reduce, the resulting update mixes gradients computed along slightly different numerical paths.
>
> However, the severity differs between pre-training and RL. In **RL training**, the mismatch directly corrupts the policy gradient signal because the loss function explicitly depends on the ratio between probabilities computed by two different engines, turning an on-policy algorithm into an implicitly off-policy one and potentially causing training collapse. In **pre-training**, the standard cross-entropy loss does not involve probability ratios between two systems, so small numerical differences tend to be averaged out rather than amplified. Prior work on low-precision training[1] and gradient quantization[2] suggests that moderate gradient noise can be tolerated during pre-training without significant convergence degradation.
>
> That said, incorporating TBIK in heterogeneous pre-training could still be beneficial in specific scenarios: (1) when debugging or reproducing training runs across different hardware configurations; (2) for models sensitive to numerical noise (e.g., early training stages or unstable loss landscapes). We believe this is a promising direction for future work and thank the reviewer for the suggestion.
>
> **Reference:**
> [1] Mixed Precision Training. ICLR2018. https://arxiv.org/abs/1710.03740
> [2] QSGD: Communication-Efficient SGD via Gradient Quantization and Encoding. NeurIPS2017. https://arxiv.org/abs/1610.02132

---

> > ### Author Rebuttal · Reviewer_sGip · 2026-04-02
> >
> > Thanks to the authors for the detailed response. I find the new RL results helpful, and believe that the clarification on how the all-gather is avoided strengthens the paper. I also appreciate the discussion regarding TP mismatch in pretraining and find it satisfactory. Therefore, I will raise the score.
> >
> > One quick question: were the timings in Figures 7 and 11 with the original all-gather-based implementation, or with the custom tree all-reduce described in W2? If former, updated timings would be very helpful.

---

> > > ### Author Response · Authors · 2026-04-03
> > >
> > > Thanks very much for your review and for the helpful suggestions. We really appreciate it, and thank you for raising the score as well.
> > >
> > > For your question, the timings in Figures 7 and 11 are based on the custom tree all-reduce, not the original all-gather implementation.

---

### Official Review · Reviewer_g8j9 · 2026-03-13

**Soundness:** 3
**Presentation:** 3
**Significance:** 3
**Originality:** 2
**Overall Recommendation:** 4
**Confidence:** 5

**Summary:**

This papers presents TBIK, which stands for Tree-Based Invariant Kernel, to solve trainining-inference precision mismatch in particularly RL training. The authors adopted tree-based computation to ensure that the computation order is not changed with varying TP sizes, leading to deterministic output under different TP sizes. The authors conducted several experiments to illustrate its determinism property with some performance degradaton. At the same time, this kind of determinism is shown to be important in the author's RL experiment where the best reward is achieved with TBIK.

**Compliance With Llm Reviewing Policy:**

Affirmed.

**Final Justification:**

This paper targets at a realistic and important problem in RL training and the authors have given a useful and simple solution to this problem.

**Key Questions For Authors:**

1. In the abstract line 22, it is mentioned that "While prior work has addressed batch-size–related nondeterminism through batch-invariant kernels, determinism across different TP sizes remains an open problem, particularly in RL settings, where the training engine typically uses Fully Sharded Data Parallel (FSDP) (i.e., TP = 1) while the rollout engine relies on multi-GPU TP to maximize the inference throughput, creating a natural mismatch between the two". This claim is questionable to me as RL frameworks such as veRL and Slime both support Megatron training backend and this seems to be quite necessary for large LLMs (e.g. larger than 30B?).
2. In the RL experiments, it would be better to show that FSDP and vLLM produce the same logits or log_probs.

**Limitations:**

yes

**Strengths And Weaknesses:**

Strength:
1. The paper is easy to follow and the concepts are clearly explained
2. The author has conducted sufficient experiments to showcase the determinism of TBIK

Weaknesses:
1. TBIK will introduce another 5% to 30% performance degradation
2. The RL experiments are conducted on 1.7B model, and experiments on larger scales will make the paper more convincing.

---

> ### Author Rebuttal · Authors · 2026-03-31
>
> ### [W1] `TBIK will introduce 5% to 30% overhead.`--We add further discussion to explain why it is acceptable in practice.
>
> We thank the reviewer for pointing this out. Here, we provide a more detailed discussion.
>
> **The practical overhead falls at the lower end of this range.** The overhead decreases with sequence length as shown in Figure11 because attention computation grows quadratically, while linear layers affected by TBIK grow only linearly. The dominant use case for TBIK, RL training of reasoning LLMs, operates in the long-sequence regime (8K–64K tokens)[1][2], where overhead is approximately 10%.
>
> **This overhead should be weighed against the cost of not having bitwise determinism.** Without TBIK, practitioners must resort to importance sampling corrections, which introduce their own computational overhead (additional logprob computation and storage) and have been shown to fail during extended training[3][4]. Training collapses can waste hours or even days of GPU time, far exceeding TBIK's steady-state overhead.
>
>
> ### [W2] `The RL experiments are limited to 1.7B model.`--We follow the same experimental setup as vLLM, and the effectiveness of TBIK is independent of model size.
>
> We appreciate the reviewer's concern. We address it from three aspects:
>
> **Established experimental setup.** Our RL experiment directly follows the setup from the official vLLM blog[5] and repository[6], which use Qwen3-1.7B on GSM8K as the reference configuration. We intentionally adopted the same model and dataset to enable direct comparison with this widely recognized baseline.
>
> **TBIK is a kernel-level contribution, independent of model size.** TBIK operates on individual MatMul and All-Reduce invocations, where the relevant parameters are the per-layer matrix dimensions and the TP configuration, not the model size. A 70B model and a 1.7B model invoke the same tree-based MatMul kernel with different matrix shapes, but the determinism guarantee is governed by the same algorithmic principles.
>
> **Extensive existing work validates that eliminating mismatch improves RL at all scales.** Much work has established that training-inference mismatch degrades RL training, and that better alignment between the two engines improves stability and performance across model sizes ranging from 0.5B to 72B[7][8]. TBIK can fundermentally guarantee bitwise-identical logprobs between two engines, enabling truly on-policy RL. The improvement is expected to generalize to larger models.
> ### [Q1] `TP seems necessary in training for large models, is TP-size mismatch a real problem?`--TP is essential for training large models but the TP size differs from that used during inference.
> We thank the reviewer for raising this question.
>
> **First, we would like to clarify a potential misconception.** FSDP and Megatron are concepts at different levels. FSDP refers to a general parallelism strategy(originally proposed as ZeRO[9]), which can also refer to PyTorch's specific implementation[10]. we use the term "FSDP" to refer to the general parallelism strategy. Megatron is a concrete training framework that supports multiple parallelism strategies including TP, PP, and FSDP itself. Also, TP is essential for training large models but not necessary. For example, Qwen2.5-32B can be trained on 8×H100 80GB GPUs using FSDP with TP = 1, since FSDP shards all model states and each GPU only stores a 1/N fraction.
>
> **Second, in practice, TP sizes are often different.** The inference side prefers a larger TP size (e.g., TP=8) to minimize per-token decoding latency, since autoregressive decoding is memory bound. The training side, by contrast, is compute-bound with large batches, so it favors a smaller TP size (e.g., TP=1/2) to avoid the frequent all-reduce communication overhead that TP introduces. Even if both stages use TP, their optimal sizes are likely to differ. Forcing the same TP size would mean at least one side operates suboptimally, leading to training inefficiency.
> ### [Q2] `Can FSDP and vLLM produce the same logits or log_probs?`--Yes, we report a zero KL divergence in Figure8.
> We thank the reviewer for the suggestion. In Figure 8, we already report the KL divergence between the probabilities produced by FSDP and vLLM. Our results show that our BIO+TBIK method achieves a KL divergence of 0, indicating two engines produce identical logits. We will further include more intuitive visualizations (e.g., direct comparisons of logits distributions) to demonstrate this consistency.
>
> **Reference:**
> [1]https://arxiv.org/abs/2501.12948
> [2]https://arxiv.org/abs/2501.12599
> [3]https://arxiv.org/abs/2512.01374
> [4]https://arxiv.org/abs/2602.10693
> [5]https://vllm.ai/blog/bitwise-consistent-train-inference
> [6]https://github.com/teja-rao/spirl?tab=readme-ov-file
> [7]https://fengyao.notion.site/off-policy-rl
> [8]https://arxiv.org/abs/2510.26788
> [9]https://arxiv.org/abs/1910.02054
> [10]https://arxiv.org/abs/2304.11277

---

> > ### Author Rebuttal · Reviewer_g8j9 · 2026-04-03
> >
> > Thank you for your clear explanation and really sorry for my late responses due to busy work these two days. I do agree with most of your points, however, just from a practical point of view, 10% overhead might be still too much when we run large-scale experiments and the easiest way to resolve the mismatch might be just to set the TP size the same. Nonetheless, I still find this work valuable as it is targeting at a major problem in RL training. Thus, I would like to raise my rating and hopefully can see your work get integrated into the existing RL frameworks such as verl and slime.

---

> > > ### Author Response · Authors · 2026-04-03
> > >
> > > We sincerely thank the reviewer for taking the time to provide valuable suggestions and feedback. We also greatly appreciate the recognition and the increased score, which means a lot to us. We would like to provide further analysis and **real-world examples to illustrate why TP sizes typically differ between training and inference, such as DeepSeek-V3[1] and Kimi K2.5[2].**
> > >
> > > **On the training side, the workload is compute-bound.** Training prefers small TP sizes (or no TP at all) because TP requires an all-reduce at every transformer layer, introducing substantial communication volume. Larger TP sizes also reduce per-GPU weight shards, lowering GPU compute utilization and making the compute-communication overlap harder to achieve. This is why many prominent models are trained with TP=1 in practice: DeepSeek-V3 [1] (Section 3.2) and Kimi K2.5 [2] (Appendix C) both explicitly document their training parallel strategies with TP=1.
> > >
> > > **On the inference side, the primary bottleneck is autoregressive decoding, which is memory-bandwidth bound.** Unlike training, decoding involves KV cache and processes only one token per step, reducing per-step computation by an order of magnitude. TP is well-suited for inference because it shards both model weights and KV cache across GPUs, directly reducing per-token decoding latency. DeepSeek-V3 [1] (Section 3.4) documents using TP=4 for inference, and vLLM provides detailed usage guides [3] recommending TP for multi-GPU inference deployment. That said, the optimal inference TP size also depends on model architecture and system configuration. For instance, DeepSeek-V3.2 [4] uses MLA with a single KV head, in which case TP requires replicating the KV cache rather than sharding it, reducing the benefit of large TP.
> > >
> > > These real-world examples demonstrate that TP size mismatch between training and inference is not a hypothetical concern but a practical reality in production LLM systems. TBIK addresses this by decoupling numerical consistency from TP size, allowing both sides to independently choose their optimal parallel strategies.
> > >
> > > Reference:
> > >
> > > [1] https://arxiv.org/abs/2412.19437
> > >
> > > [2] https://arxiv.org/abs/2602.02276
> > >
> > > [3] https://docs.vllm.ai/projects/recipes/en/latest/DeepSeek/DeepSeek-V3.html
> > >
> > > [4] https://docs.vllm.ai/projects/recipes/en/latest/DeepSeek/DeepSeek-V3_2.html#installing-vllm

---

### Official Review · Reviewer_qbgJ · 2026-03-14

**Soundness:** 3
**Presentation:** 4
**Significance:** 3
**Originality:** 3
**Overall Recommendation:** 5
**Confidence:** 4

**Summary:**

The paper first explains the problems with current batch-invariant computations in guaranteeing determinism across different parallelization schemes at both the inter- and intra-GPU levels. Next, it introduces a binary-tree-based accumulation strategy at both the intra- and inter-GPU level to achieve full determinism in the accumulation order. The results show that, across different tensor parallelism factors, the proposed computation strategy indeed achieves similar results with tolerable overhead.

**Compliance With Llm Reviewing Policy:**

Affirmed.

**Final Justification:**

The rebuttal resolved my issues and my sentiment regarding the paper remains the same.

**Key Questions For Authors:**

The paper explores an important topic: achieving determinism in computations, especially for reasoning-based LLMs that rely on RL. I really liked the observations and the holistic characterization of how non-determinism is introduced into the accumulation order of contraction operators, such as MatMul. This is pedagogical and also shows that batch-invariant computations address only one aspect of determinism.

The proposed technique is simple, yet effective. The tree-based reduction using binary trees supports arbitrary tensor parallelism factors while guaranteeing a single accumulation order. The results corroborate that this is indeed the case. A theorem statement on this aspect would have been more effective to solidify the fact that the accumulation order is invariant across different intra- and inter-GPU settings. Overall, I find the idea to be neat.

The authors could have elaborated more on the overheads, though, at a theoretical level. Currently, overheads are measured and are tolerable, but the paper does not provide an understanding of them. For example, what would happen if you cannot create a balanced binary tree? Will that affect load balancing, etc.? What if there are stragglers in GPU computations, and what freedoms do we have to move computations around? Some more in-depth conceptual discussion is needed to make the discussion about overheads complete.

Overall, I like this work, and I believe it would have an impact on the design of deterministic kernels.

**Limitations:**

Yes

**Strengths And Weaknesses:**

Strengths
* Good obervations that wholistically characterize the non-determinism involved in the computations
* A simple, yet effective strategy to introduce determinism to accumulations at both intra- and inter-GPU levels
* Strong determinism related results with tolerable overhead

Weaknesses
* The limitations of load balancing issues are not discussed fully
* The authors could have optimized the kernels according to what is discussed in the evaluation section.

---

> ### Author Rebuttal · Authors · 2026-03-31
>
> ### [W1&Q1]`The theoretical analysis of the overhead is relatively limited, including overhead decomposition, load balancing, and stragglers.`--We provide a more detailed discussion of the overhead from these three aspects.
> We thank the reviewer for this insightful question. We will discuss this from three aspects:
>
> **1. Theoretical overhead decomposition**
> The overhead of TBIK comes from three components: (1) **loss of fused accumulation**: replacing Tensor Core's fused MMA accumulate with explicit pairwise additions. The number of additions is identical (`TILE_K - 1`), but they cannot leverage the hardware accumulate path; (2) **register pressure**: maintaining multiple intermediate accumulators for the tree levels increases per-thread register usage from ~82 to ~178 registers, approaching the hardware limit of 255 and potentially causing register spilling; (3) **control flow overhead**: the tree traversal logic requires O(1) amortized comparisons per K-tile. Among these, component (1) and (2) are the dominant sources, while (3) is negligible.
>
> **2. Load balancing**
>
> **Unbalanced trees.** Our implementation handles non-power-of-2 TILE_K through the `FIRST_LEVEL_BLOCK` parameter, which allows the bottom level of the tree to have a fan-in greater than 2. Accumulation within this level still follows a fixed sequential order, preserving determinism. The overhead impact is minimal, only affecting the constant factor, not asymptotic behavior. **Intra-GPU load balancing.** Our persistent kernel assigns output tiles to SMs uniformly, and all tiles perform the same number of K-tile iterations. The tree reduction logic has data-dependent branches (carry vs. no-carry), but these are coherent across all threads within a warp. All threads process the same K-tile index and take the same branch, so no warp divergence occurs. **Cross-GPU load balancing.** In practice, TP partitions the K dimension evenly across GPUs (K must be divisible by TP_size), resulting in balanced workloads. If K is not evenly divisible, padding can be applied, which is a standard practice in existing TP implementations.
>
> **3. Straggler mitigation and degrees of freedom**
>
> The reviewer correctly notes that our fixed tree topology limits dynamic reordering. We make three observations: First, **this constraint is inherent to any deterministic reduction scheme**. Determinism requires a fixed execution order, trading scheduling flexibility for reproducibility. This is the fundamental tradeoff, not a limitation specific to TBIK. Second, **the straggler problem is largely orthogonal to TBIK.** GPU stragglers affect all collective operations (including standard NCCL all-reduce), and mitigation strategies (redundant computation, asynchronous pipelines, straggler-aware scheduling) operate at the system level, above the kernel level where TBIK operates. TBIK does not preclude these optimizations. Third, **independent subtrees within the tree structure can still be scheduled concurrently.** For example, in a 4-GPU tree, GPUs 0+1 and GPUs 2+3 perform their pairwise reductions in parallel.
> ### [W2] `The kernel performance optimization is not sufficiently thorough.`--We want to clarify that the suboptimal-performance kernels are only applied to two layers, which is provably sufficient to guarantee bitwise-identical results as verified by both theoretical analysis and empirical evaluation. The resulting end-to-end overhead is limited and acceptable. We will also further optimize the kernels on more advanced GPUs.
> We thank the reviewer for raising this point. We would like to provide context on why the end-to-end impact is limited and acceptable.
>
> **TBIK is only applied to two layers per transformer block.** Through careful analysis, we identified that TP-induced non-determinism originates exclusively from the row-parallel linear layers: `o_proj` in attention and `down_proj` in FFN. Applying TBIK to just these two layers is sufficient to achieve bitwise-identical results across TP sizes. All other layers remain unmodified and run at full cuBLAS speed, resulting in 5–30% end-to-end overhead as shown in Figure7.
>
> **The practical overhead falls at the lower end of this range.** The overhead decreases with sequence length as shown in Figure11 because attention computation grows quadratically O(n²), while linear layers affected by TBIK grow only linearly O(n). The dominant use case for TBIK, RL training of reasoning LLMs, operates in the long-sequence regime (8K–64K tokens), where overhead is approximately 10%.
>
> **Our development was constrained by hardware availability.** Primary development was conducted on L40s workstations (Ada Lovelace, SM89), which lack key Hopper features such as Warp Specialization and TMA, two of the most impactful optimizations. We are also actively exploring architecture-agnostic optimizations such as autotuning for block sizes. We will incorporate these improvements and report updated results in the revised version.

---

> > ### Author Rebuttal · Reviewer_qbgJ · 2026-04-03
> >
> > Good explanation. I still like this paper, and it provides solid contributions!

---

### Decision · Program_Chairs · 2026-04-30

**Decision:**

Accept (regular)

**Comment:**

The paper proposes a novel approach (Tree-Based Invariant Kernels) to eliminate the nondeterminism of kernel computation with different TP size, which is very important for improving the stability of large scale RL training. The motivation is straightforward, and the experiments are quite solid that clearly show the benefit. Reviewers do have some concerns (e.g., limited experiments, limited scope and potential slowdown if used in the real-world scenarios), which are valid. Authors address these concerns well. Overall AC still thinks the work gives a meaningful contribution.